# Numerical Simulation and Structural Optimization of Swirl Flow Micro-Nano Bubble Generator

Xinkang Hu [1], Bo Zhang [2,3,*], Chundu Wu [1], Xiaohong Xu [2], Mingming Xue [2] and Xiaoyong Zheng [4]

1 School of Agricultural Engineering, Jiangsu University, Zhenjiang 212000, China; huxk@stmail.ujs.edu.cn (X.H.); wcdujs@126.com (C.W.)
2 School of the Environment and Safety Engineering, Jiangsu University, Zhenjiang 212000, China; xuxiaohong@yeah.net (X.X.); xuemm0406@163.com (M.X.)
3 Changzhou Engineering and Technology Institute of Jiangsu University, Changzhou 213164, China
4 Jiangsu Huanchuan Environmental Engineering Co., Ltd., Taizhou 225300, China; ozonemeta@126.com
* Correspondence: tabol@126.com; Tel.: +86-138-1515-9720

**Abstract:** The development of the bubble generator that can efficiently generate micro-nano bubbles has always been recognized as a challenge. Swirling flow is considered to be an efficient method to enhance hydrodynamic cavitation. The vortex supply chamber and the variable-diameter accelerated vortex cavitation reaction chamber were combined to obtain a stable high-speed tangential liquid flow and improve the cavitation effect inside the generator in this study. The central air intake column was innovatively installed above the cavitation reaction chamber, which prolonged the shear fracture time of bubbles under high shear force and improved the gas–liquid contact and mixing efficiency. The influence of geometric parameters on the internal and external flow fields of the generator was analyzed through the numerical simulation. The optimized central air intake column was located 10 mm above the inlet of the cavitation reaction chamber. The optimized variable diameter contraction angle was 16°, and the optimized generator outlet diameter was 15 mm. Through the bubble performance test, it was verified that the micro-nano bubbles with the minimum size and average size of 0.31 μm and 3.42 μm could be generated by the manufactured generator. The enforcement of the research provided theoretical guidance and data support for the development of efficient micro-nano bubble generators.

**Keywords:** micro-nano bubbles; numerical simulation; swirl flow; bubble size; multiphase flow

## 1. Introduction

Micro-nano bubbles refer to the bubbles with a diameter of less than 100 μm [1]. Because of the unique physical properties, such as small diameter, large specific surface area, stable phase interface, and long residence time in the liquid [2–4], they have become a significant improvement measure to improve mass transfer efficiency [5] and reduce operating costs. They are widely used in water treatment [6], nanofluid therapy [7,8], mineral flotation [9,10], and plant growth promotion [11].

In the past few decades, several typical micro-nano bubble generators based on hydrodynamic cavitation have been developed, such as the pressurized dissolution type [12], Venturi type [13], and ejector type [14]. Among them, the swirl flow type has received extensive attention in recent years, due to its simple structure, high efficiency of bubble generation, and low energy consumption. Meanwhile, the size of the micro-nano bubbles and the particle size distribution of bubbles are important indicators for evaluating the performance of micro-nano bubble generators, which significantly affect the mass transfer between gas and liquid phases [15]. Therefore, the use of the swirl flow field to increase the turbulence level of flowing liquid, enhance the shear and fracture of bubbles in the flow state, and reduce the diameter of bubbles generated in the micro-nano bubble generator has become a hot spot in current related research.

In terms of improving the vortex microbubble generator, researchers have made many attempts. Wu et al. [16] designed a baffle microbubble generator and reported that the change in bubble size was more sensitive to liquid flow than to gas phase. Ding et al. [17] designed an axial vortex microbubble generator to promote bubble rupture by installing a new bubble rupture device, which greatly reduced the size of the bubbles generated by traditional Venturi tubes. Kogawa et al. [18] used the Coanda effect to increase the level of vortex breakdown and reduced the size of the generated bubbles by improving the outlet structure of the vortex microbubble generator. In our previous work, we designed a traditional swirl cavitation micro-nano bubble generator [19]. The minimum diameter of the generated micro-nano bubbles was 16.257 μm. Although the swirling flow caused strong gas self-priming ability, the size results of bubble generation were still unsatisfactory. The reason was that the simple structure brought additional local flow and pressure loss, resulting in greater energy consumption. In terms of numerical simulation methods, the Euler–Euler multiphase flow model is considered to be a good choice to simulate the interaction between gas and liquid phases in the generator [20].

In this work, a new type of swirling micro-nano bubble generation strategy was proposed. The vortex supply chamber and the variable-diameter accelerated vortex cavitation reaction chamber were combined. The high-speed liquid phase fluid was pumped into the high-speed liquid phase fluid through the inlet pipe tangentially connected to the swirling chamber, and the vortex was formed inside the swirling cylinder [21]. The vortex center produced a stable negative pressure axis to inhale air and spray out of the swirling cylinder with the liquid phase to generate micro-nano bubbles. While obtaining a stable high-speed tangential flow, the cavitation effect inside the generator was improved. The central air inlet column was innovatively installed above the cavitation reaction chamber, which prolonged the shear fracture time of the bubbles under high shear force and improved the gas–liquid contact and mixing efficiency. Through the numerical simulation results, the influence of geometric parameters (central air intake column, variable diameter contraction angle, and outlet diameter) on the velocity, turbulent kinetic energy, and turbulent dissipation rate in the internal and external flow fields of the generator was analyzed. Then, the swirl flow micro-nano bubble generator was manufactured, and the performance test of the micro-nano bubble generator was carried out. The performance was comprehensively evaluated according to the bubble size and the particle size distribution of the bubble.

## 2. Materials and Methods

### 2.1. Geometrical Model

The geometrical structure of the micro-nano bubble generator is shown in Figure 1. It consisted of an eddy current supply cavity of the outer ring and a variable-diameter accelerated eddy current cavitation reaction cavity of the inner ring.

The liquid was continuously pumped into the vortex supply cavity of the outer ring of the generator through the liquid inlet at a high speed so that the liquid flowed circumferentially and formed a stable vortex in the outer cavity. With the continuous supply of liquid, the liquid flow entered the inner ring reaction chamber tangentially after returning through the baffle plate, forming an accelerated eddy current in the reaction chamber. The negative pressure was generated at the center of the eddy current, and the air was inhaled by the gas inlet, which broke into the initial bubble under the action of the flowing liquid.

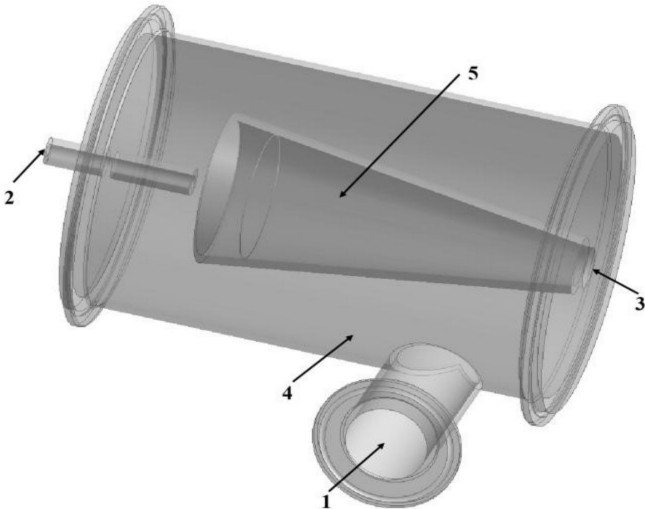

**Figure 1.** Schematic diagram of the swirl flow micro-nano bubble generator. Annotation: 1. water inlet; 2. air inlet; 3. outlet; 4. vortex supply cavity; 5. variable-diameter-accelerated eddy current cavitation reaction cavity.

In order to further reduce the size of the bubble, a variable-diameter accelerated vortex cavitation reaction chamber was added to the inner ring to improve the shear fracture effect of the bubble. The gas–liquid mixture entered the variable-diameter accelerated vortex cavitation reaction chamber, and the turbulence caused by the flowing liquid was sheared into several discrete micro-nano bubbles. The geometric parameters of the micro-nano bubble generator are shown in Table 1, and the sectional view is shown in Figure 2.

**Table 1.** The geometric parameters of the micro-nano bubble generator.

| Generator Diameter | Generator Length | Generator Angle |
|---|---|---|
| $d_1 = 3$ mm | $L_1 = +10, 0, -10$ mm | $\theta = 12, 14, 16, 18°$ |
| $d_2 = 25$ mm | $L_2 = 125$ mm | - |
| $d_3 = 13, 15, 17$ mm | $L_3 = 155$ mm | - |
| D = 100 mm | - | - |

'+' represents the air inlet end located above the cavitation reaction chamber; '−' represents below.

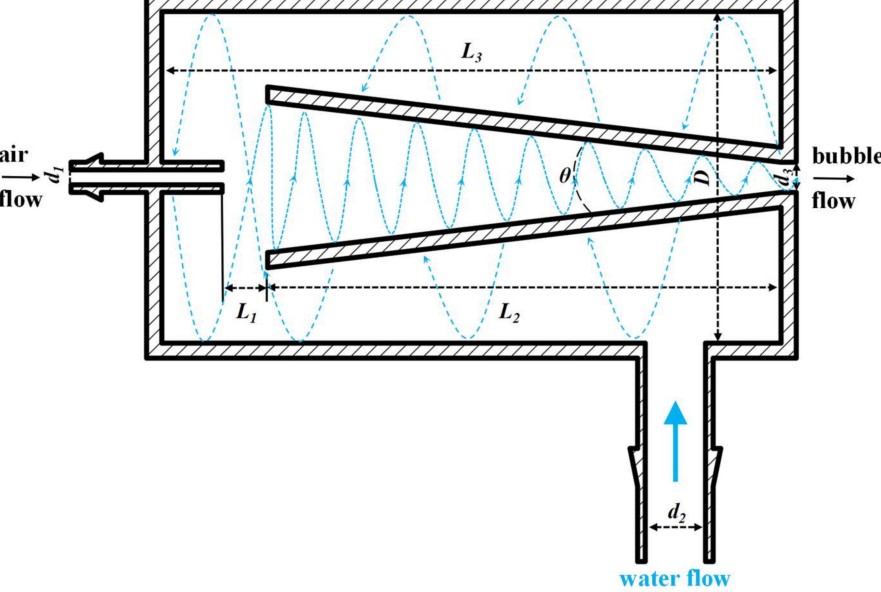

**Figure 2.** The sectional view of the micro-nano bubble generator.

### 2.2. Multiphase Model and Turbulence Model

The Euler–Euler multiphase flow model was used to simulate the interaction between air and liquid phases in the internal flow field. Each phase was viewed as a mutually penetrating continuum and depended on the ensemble mean of the multiphase Navier–Stokes (RANS) equations. In addition to considering the swirling effect, the derivative equation of effective viscosity was also considered for the realizable k-epsilon ($k$-$\varepsilon$) model. Therefore, it could calculate low Reynolds number turbulence. The mass equation and momentum conservation equation are expressed as follows:

Continuity equation:

$$\frac{\partial \alpha_q}{\partial t} + \nabla \cdot \left( \alpha_q \vec{v}_q \right) = 0 \tag{1}$$

Mass equation:

$$\frac{\partial}{\partial t} \left( \alpha_q \rho_q \right) + \nabla \cdot \left( \alpha_q \rho_q \vec{v} \right) = \sum_{p=1}^{n} \left( \dot{m}_{pq} - \dot{m}_{qp} \right) + S_q \tag{2}$$

Momentum equations:

$$\frac{\partial}{\partial t} \left( \alpha_q \rho_q \vec{v}_q \right) + \nabla \cdot \left( \alpha_q \rho_q \vec{v}_p \vec{v}_q \right) = -\alpha_q \nabla p + \nabla \cdot \overline{\tau_q} + \alpha_q \rho_q g \\ + \sum_{p=1}^{n} \left( R_{pq} + \dot{m}_{pq} \vec{v}_{qp} - \dot{m}_{qp} \vec{v}_{qp} \right) + \alpha_q \rho_q \left( F_q + F_{lift,q} + F_{vm,q} \right) \tag{3}$$

Phase stress–strain tensor:

$$\left[ \overline{\tau_q} \right] = \alpha_q \mu_q \left( \nabla \vec{v}_q + \nabla \vec{v}_q^{-T} \right) + \alpha_q \left( \lambda_q - \frac{2}{3} \mu_q \right) \nabla \cdot \vec{v}_q [I] \tag{4}$$

Reynolds stress tensor:

$$\tau_q' = -\frac{2}{3} \left( \rho_q k_q + \rho_q \mu_{t,q} + \nabla \vec{v}_q [I] \right) + \rho_q \mu_{t,q} \left( \nabla \vec{v}_q + \nabla \vec{v}_q^{-T} \right) \tag{5}$$

Turbulent viscosity:

$$\mu_{t,q} = \rho_q C_\mu \frac{k_q^2}{\varepsilon} \tag{6}$$

The realizable $k$-$\varepsilon$ model was used for numerical calculation, considering computational complexity and accuracy. The turbulence equation and the turbulence dissipation rate equations are as follows:

Turbulence equation $k_q$:

$$\frac{\partial}{\partial t} \left( \alpha_q \rho_q k_q \right) + \nabla \cdot \left( \alpha_q \rho_q \vec{v}_p k_q \right) = \nabla \cdot \left( \alpha_q \frac{\mu_{t,q}}{\sigma_k} \nabla k_q \right) + \left( \alpha_q G_{k,q} - \alpha_q \rho_q \varepsilon_q \right) \\ + \sum_{l=1}^{N} K_{lq} \left( C_{lq} - C_{ql} k_q \right) - \sum_{l=1}^{N} K_{lq} \left( \vec{v}_l - \vec{v}_q \right) \frac{\mu_{t,l}}{\alpha \sigma} \nabla \alpha_l + \sum_{l=1}^{N} K_{lq} \left( \vec{v}_l - \vec{v}_q \right) \frac{\mu_{t,l}}{\alpha \sigma} \nabla \alpha_q \tag{7}$$

Turbulence dissipation rate $\varepsilon_q$:

$$\frac{\partial}{\partial t} \left( \alpha_q \rho_q \varepsilon_q \right) + \nabla \cdot \left( \alpha_q \rho_q \vec{v}_p \varepsilon_q \right) = \nabla \cdot \left( \alpha_q \frac{\mu_{t,q}}{\sigma_k} \nabla \varepsilon_q \right) + \frac{\varepsilon_q}{k_q} \left[ C_{1\varepsilon} \alpha_q G_{kq} - C_{2\varepsilon} \alpha_q \rho_q \varepsilon_q \right. \\ \left. + C_{3\varepsilon} \left( \sum_{l=1}^{N} K_{lq} \left( C_{lq} - C_{ql} k_q \right) - \sum_{l=1}^{N} K_{lq} \left( \vec{v}_l - \vec{v}_q \right) \frac{\mu_{t,l}}{\alpha \sigma} \nabla \alpha_l + \sum_{l=1}^{N} K_{lq} \left( \vec{v}_l - \vec{v}_q \right) \frac{\mu_{t,l}}{\alpha \sigma} \nabla \alpha_q \right) \right] \tag{8}$$

where the constants have the values of: $C_{1\varepsilon} = 1.44$, $C_{2\varepsilon} = 1.92$, and $C_{3\varepsilon} = 0.09$.

### 2.3. Mesh Generation

In this work, ANSYS Meshing was used for mesh generating, and Computational Fluid Dynamics (CFD) software ANSYS Fluent 2022R1 was used for all simulations. Figure 3

shows the mesh of the swirl flow micro-nano bubble generator and outflow field. In order to ensure high grid quality, tetrahedral-densified grids were used in local complex computational domains.

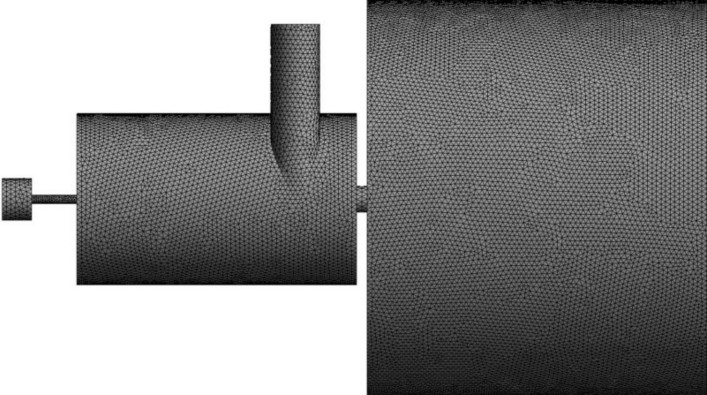

**Figure 3.** Mesh of the swirl flow micro-nano bubble generator and outflow field.

Through the mesh independence verification (Table 2), with the increase in the number of grids, the calculation error of the outlet velocity and the outlet turbulent kinetic energy gradually decreased. When the number of grids increased from 2,986,759 to 4,153,679, it tended to be stable, and the error decreased to less than 1%. At this time, it could be considered that the calculation results had little to do with the number of grids. The unit quality was 0.855. The aspect ratio and skewness of the mesh were 1.78 and 0.2, respectively, and the size of a typical grid element was 3 mm. A total of 2,986,759 tetrahedral elements and 521,689 nodes were used in the simulation.

**Table 2.** Grid independence verification.

| Grid Number | Outlet Velocity (m/s) | Outlet Turbulent Kinetic Energy (m$^2$/s$^2$) |
| --- | --- | --- |
| 1,342,368 | 26.83 | 43.82 |
| 2,087,354 | 28.74 | 46.73 |
| 2,986,759 | 29.34 | 48.51 |
| 4,153,679 | 29.56 | 48.48 |

*2.4. Boundary Conditions and Numerical Setting*

In setting boundary conditions, the pressure-based, transient-state, and absolute velocity formula was adopted in the simulation, with the acceleration of gravity normal parallel to the shaft of the air inlet and outlet (*z*-axis) being 9.81. The transient simulation was selected as the analysis type, the total time was 1 s, and the time step was 0.001 s. In the inlet section, according to the design conditions, the inlet mass flow rate of the liquid was set to 15 m$^3$/h. In the air inlet, since the micro-nano bubble generator was designed as a self-inspirator, the air inlet pressure was set to 1.0 atm. In addition, since the head pressure was less than 1 m, the pressure of the outlet was set to 1.0 atm. Gas–liquid two-phase flow was considered to be incompressible, three-dimensional, and unsteady. The continuous phase was water, with a density of 998.2 kg/m$^3$ and a viscosity of $1.0 \times 10^{-3}$ kg/m·s. The discrete phase was air, with a density of 1.225 kg/m$^3$ and a viscosity of $1.8 \times 10^{-5}$ kg/m·s. In the control equation of Section 2.2, the standard *k-ε* turbulence model for the two-phase flow simulation was selected by using the second-order upwind discretization scheme, turbulent kinetic energy, and turbulent dissipation ratio. The heat transfer process was a homogeneous model and isothermal.

*2.5. Bubble Size Measurement*

The swirl flow micro-nano bubble generator was an integrated body composed of a vortex supply chamber in the outer ring and a variable-diameter accelerated vortex

cavitation reaction chamber in the inner ring, which was made of 304 stainless steel. The experimental device included a 20 L acrylic tank, a diaphragm pump, a swirl flow micro-nano bubble generator, a pressure gauge, a valve, a gas flowmeter, and a liquid flowmeter (see Figure 4). Under the hydraulic drive of the diaphragm pump, the water in the tank entered the micro-nano bubble generator through the valve, the flowmeter pressure gauge, and finally returned to the tank for recycling. Air was automatically sucked into the water flow through the gas inlet. In order to evaluate the performance of the micro-nano bubble generator more intuitively, a high-power electron microscope was used to collect the shape image of the micro-nano bubble. The bubble image was processed by matlab, binarization, bubble feature extraction, histogram drawing, and other steps to record the size of the micro-nano bubble. During the test, a certain volume of tap water was placed into a large cylindrical acrylic tank. The liquid flow rate was 15 m$^3$/h. The micro-nano bubbles generated by the previously optimized bubble generator were directly released into the outer watershed outside of the generator. After 5 min of continuous operation, a syringe was used to draw bubbles from the middle of the acrylic tank. Then, the water droplets containing bubbles were placed on the glass slide, and the other side of the glass slide was attached to a scale of 10 μm per unit length. They were placed together under the microscope to measure the sizes of the bubbles.

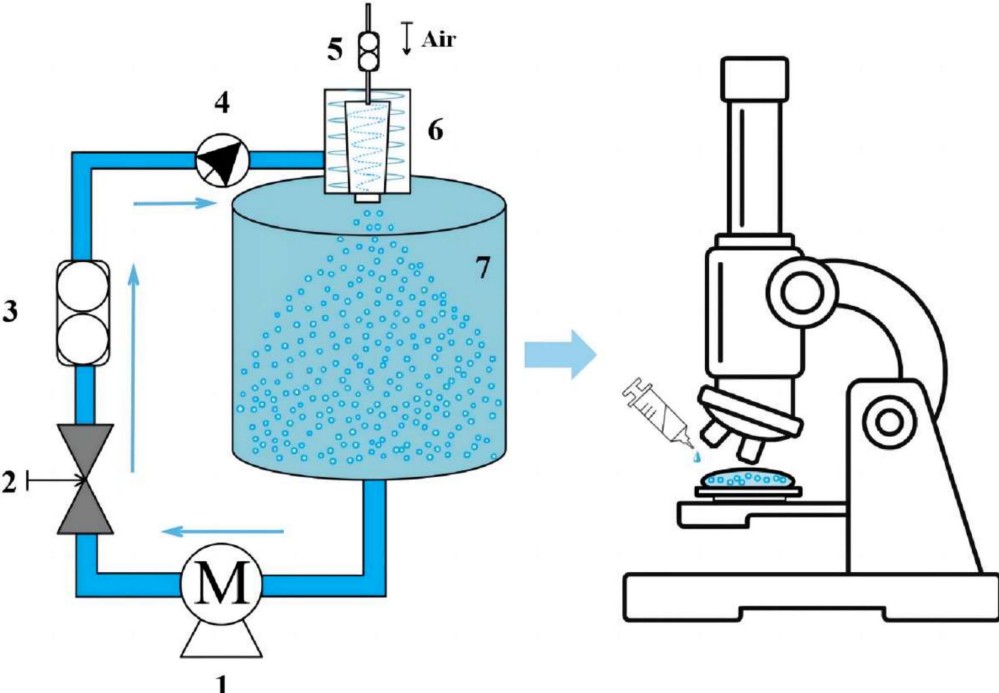

**Figure 4.** The schematic diagram of the construction of the experimental device. Annotation: 1. diaphragm pump; 2. valve; 3. liquid flowmeter; 4. pressure gauge; 5. gas flowmeter; 6. micro-nano bubble generator; 7. acrylic tank.

## 3. Results and Discussion

### 3.1. Effect of Central Air Intake Column on Bubble Generation

The inlet position ($L_1$) of the central intake column had a significant effect on the flow field structure and bubble size in the generator [22]. The relative position relationship between the central intake column and the variable-diameter accelerated vortex cavitation reaction chamber directly affected the liquid turbulence level in the cavitation reaction chamber, which can promote bubble breakup. According to Lasheras et al. [23], the diameter $d$ of the micro-nano bubbles generated under the turbulence dissipation rate $\varepsilon$ is related as follows:

$$d = C_4 \left( \frac{\sigma^3}{\rho^3 \varepsilon^2} \right)^{0.2} \tag{9}$$

In this study, we designed the intake positions of three intake columns (see Figure 5), which were located at the upper, middle, and lower parts of the inlet of the vortex cavitation reaction chamber ($L_1$ = +10 mm, 0 mm, −10 mm).

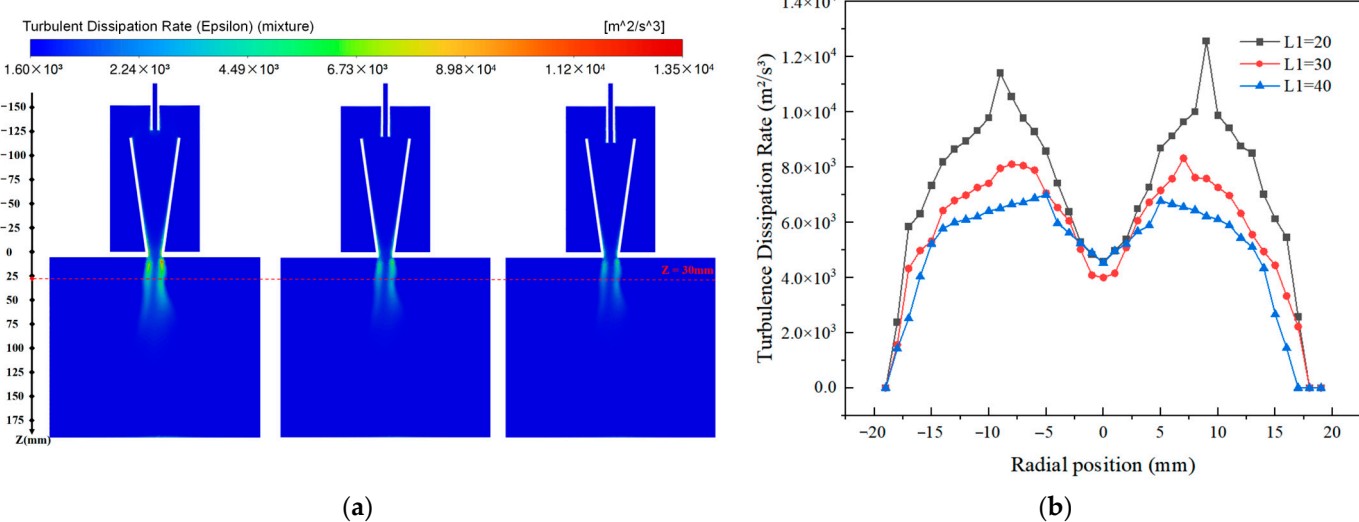

(**a**)                                                                   (**b**)

**Figure 5.** Turbulent kinetic energy distribution cloud. (**a**) Turbulent dissipation rate distribution curves (**b**) at the cross-section Z = 30 mm.

The results showed that the inlet end was located at 10 mm above the cavitation reaction chamber, which was in the flow field of the generator with $L_1$ = +10 mm, due to the longer flow distance and stronger shear force between the fluids. The turbulent dissipation rate of the flow field was higher, and the distribution range was wider. Combined with the research conclusion of Lasheras [24], the bubble size was inversely proportional to the turbulent dissipation rate. When $L_1$ = +10 mm, the turbulent dissipation rate at this time was 13,486.7 $m^2/s^3$, and the bubble diameter at the maximum turbulent kinetic energy was calculated to be 0.37 μm by Equation (9). We can infer that the inlet end located at 10 mm above the cavitation reaction chamber can be used as an ideal choice for the design of the central inlet column.

### 3.2. Effect of Variable Diameter Contraction Angle on Bubble Generation

Once the gas–liquid mixture enters the accelerated vortex cavitation reaction chamber, it will shear to form several independent fluids under the action of swirling flow. Under the acceleration guidance of the variable diameter contraction angle (θ), each independent fluid was injected into the outer basin from the outlet of the generator at a certain tangential angle to form a fluid mixture. In order to obtain the optimal structural parameters of the cavitation reactor, it was necessary to analyze the influence of the variable diameter contraction angle on the flow field distribution. The tangential velocity distribution curve at the cross-section Z = 0 mm is shown in the Figure 6. Due to the design of the variable diameter contraction angle, the fluid velocity in the vortex cavitation reaction chamber had a considerable part of the tangential component in addition to the axial component, and the tangential velocity distribution curve was anti-symmetrical. When the variable diameter contraction angle was designed to be 14, 16, and 18°, the tangential component at the outlet was more prominent, forming an axial vortex state. The independent fluids in the cavitation reaction chamber did not collide with each other during mixing, and strong shear force was generated, which was more conducive to the generation of small-sized bubbles.

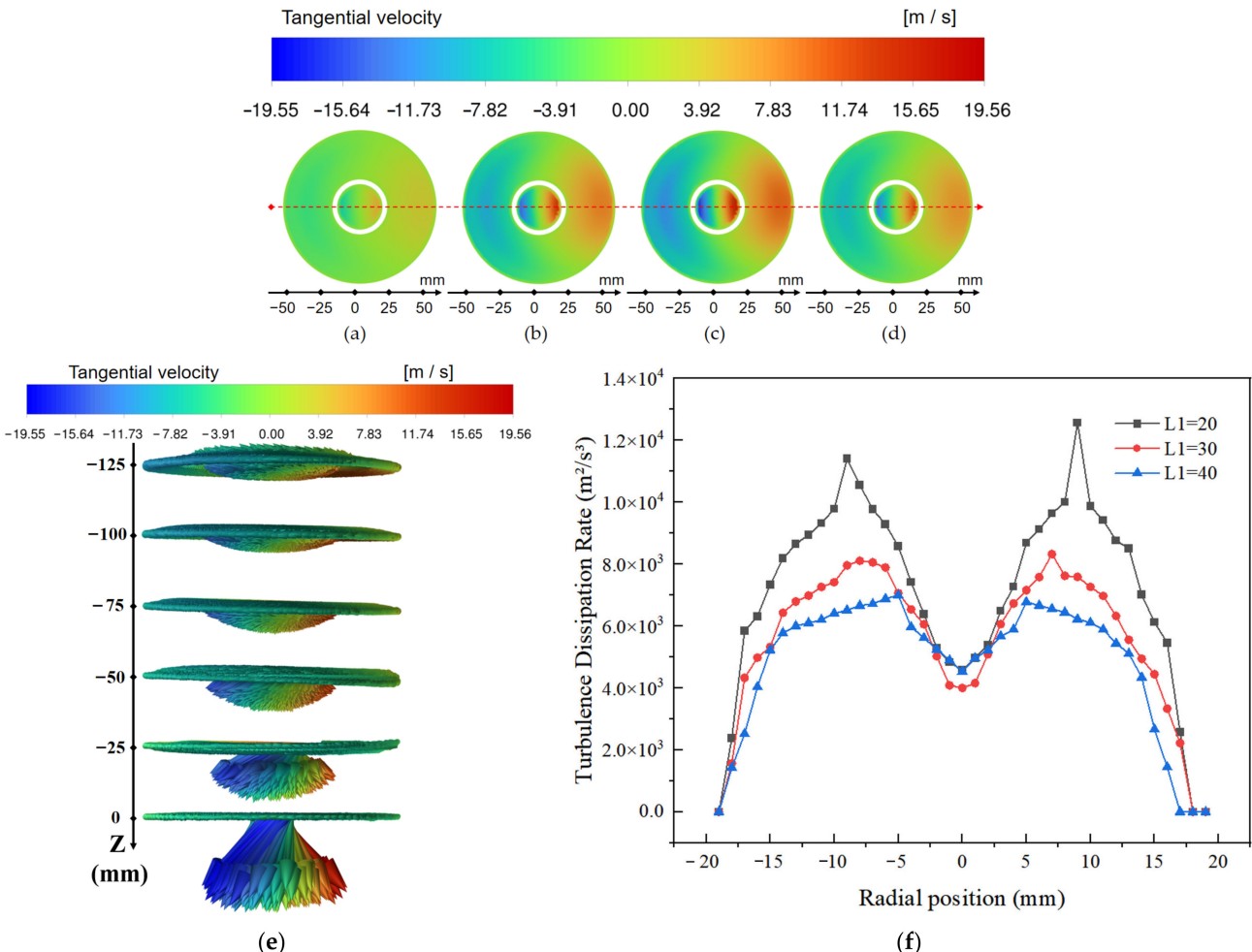

**Figure 6.** Tangential velocity distribution cloud counter (θ = 12° (**a**), θ = 14° (**b**), θ = 16° (**c**), and θ = 18° (**d**)). Tangential velocity vector θ = 16° (**e**) and tangential velocity distribution curves (**f**) at the cross-section Z = 0 mm.

The turbulent kinetic energy distribution of the internal and external flow fields of the tangential vortex micro-nano bubble generator is shown in Figure 7. When θ was 12° and 18°, the maximum turbulent kinetic energies of the external flow field of the generator were 37.48 $m^2/s^2$ (θ = 12°) and 57.83 $m^2/s^2$ (θ = 18°). The turbulent kinetic energy distribution was obviously biased to one side (θ = 18°), which led to the occurrence of polarization [25] and seriously affected the bubble size distribution at the outlet. This was in sharp contrast to the flow field inside and outside the generator when the θ was 14° and 16°. At this time, the maximum turbulent kinetic energies of the external flow field of the generator were 61.73 $m^2/s^2$ (θ = 14°) and 60.51 $m^2/s^2$ (θ = 16°). The turbulent kinetic energy distributions in the flow field inside and outside the generator were more uniform. When θ was 16°, the strong swirl significantly enhanced the turbulent kinetic energy of the mixed phase, the turbulent kinetic energy distribution in the variable-diameter accelerated vortex cavitation reaction chamber was average, and the minimum value was 43.91 $m^2/s^2$, which was much higher than the others. The instability of the gas–liquid interface increased, the surface tension of the bubble decreased, and it was easier to break into the micro-nano level.

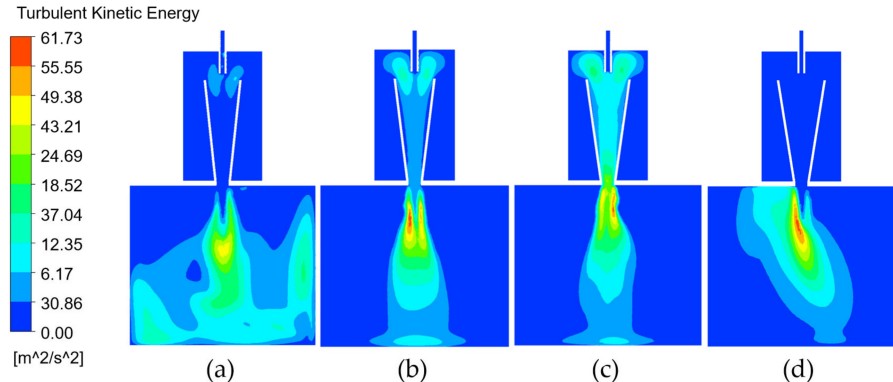

**Figure 7.** Turbulent kinetic energy cloud diagram at different angles (θ). Annotation: (**a**). θ = 12°; (**b**). θ = 14°; (**c**). θ = 16°; and (**d**). θ = 18°.

### 3.3. Effect of Outlet Diameter on Bubble Generation

The velocity contours of different cross-sections along the flow direction are shown in Figure 8. In the vortex supply chamber of the outer ring, with the continuous pumping of the liquid flow, the liquid flow gradually formed a uniform annular velocity gradient from the liquid flow inlet to the upper baffle plate, indicating that a stable vortex was gradually formed in the vortex supply chamber of the outer ring, which provided a good velocity input for the subsequent tangential acceleration of the liquid flow. In the variable-diameter accelerated vortex cavitation reaction chamber of the inner ring, the liquid flow velocity gradually increased with the decrease in the cross-sectional area of the flow channel and reached the maximum at the outlet (see Figure 8).

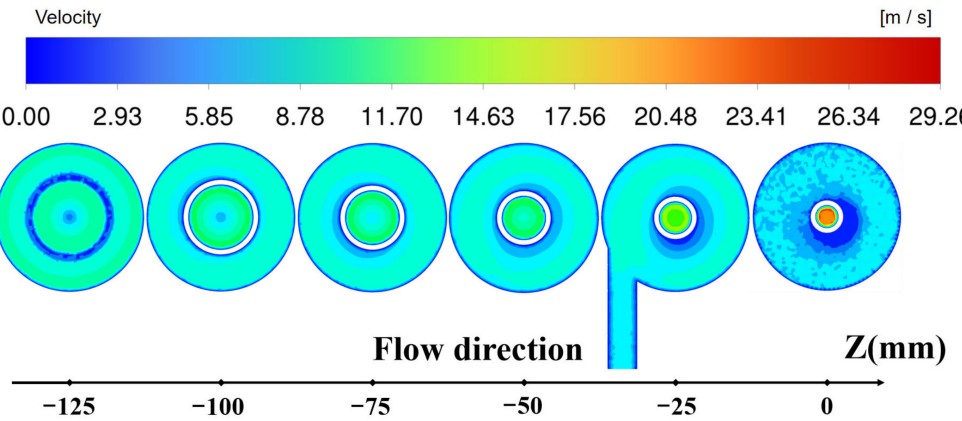

**Figure 8.** Velocity clouds of different cross-sections along the flow direction (θ = 16°).

According to the experimental results of Fujiwara et al. [26], the diameter d of the micro-nano bubbles generated at the outlet was inversely proportional to the liquid flow velocity ($v_{outlet}$) at the outlet of the generator, as shown in Equation (10). That is, the larger the liquid flow velocity at the outlet, the smaller the bubble diameter. The outlet velocity clouds with different outlet diameters are shown in Figure 9. Comparing the outlet velocity of different outlet diameters ($d_3$), the outlet with a diameter of 15 mm was the superior design choice.

$$d = 5.47 v_{outlet}^{-1.0} \tag{10}$$

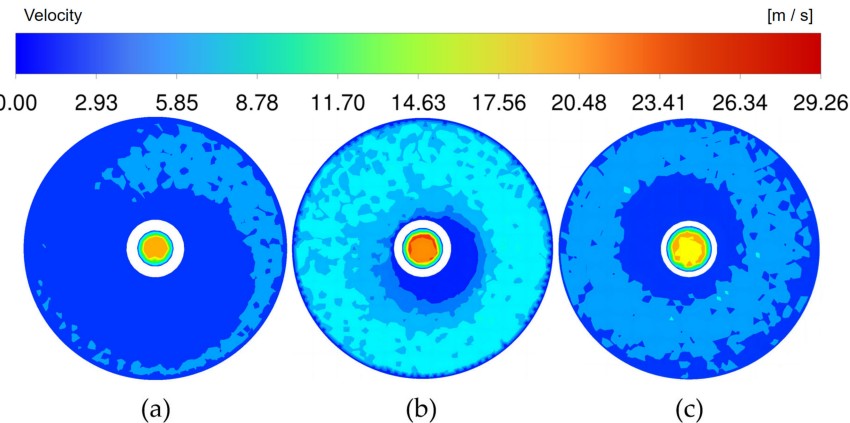

**Figure 9.** Outlet velocity clouds with different outlet diameters ($d_3$) Z = 0 mm. Annotation: (**a**). $d_3$ = 13 mm; (**b**). $d_3$ = 15 mm; (**c**). $d_3$ = 17 mm.

The pressure inside the generator has always been considered a measure of whether the cavitation effect is strong or not in the numerical simulation experiment. The positive pressure can make the medium molecules become closer to each other, and the negative pressure makes the medium molecules pull apart and become sparse. When the negative pressure reaches a certain degree, it forms a force that can pull apart the molecules of the mixed phase, which will cause structural damage to the mixed medium, resulting in the contraction or even collapse of bubbles, forming a cavitation effect [27]. The axial static pressure distribution curve of the variable-diameter accelerated vortex cavitation reaction chamber of the inner ring of the generator is shown in Figure 10. The static pressure distribution law formed by different outlet diameters was basically the same. The axial static pressure began to decrease from the inlet of the cavitation reaction chamber and showed a decreasing trend along the direction of fluid flow. The minimum value appeared at the generator outlet. When the outlet diameter was 15 mm, the minimum static pressure at the outlet was the lowest, which was $-1.644 \times 10^5$ Pa. Under this condition, the cavitation formation, development, and collapse of the gas inside the liquid were intensified, and the cavitation effect was the most significant.

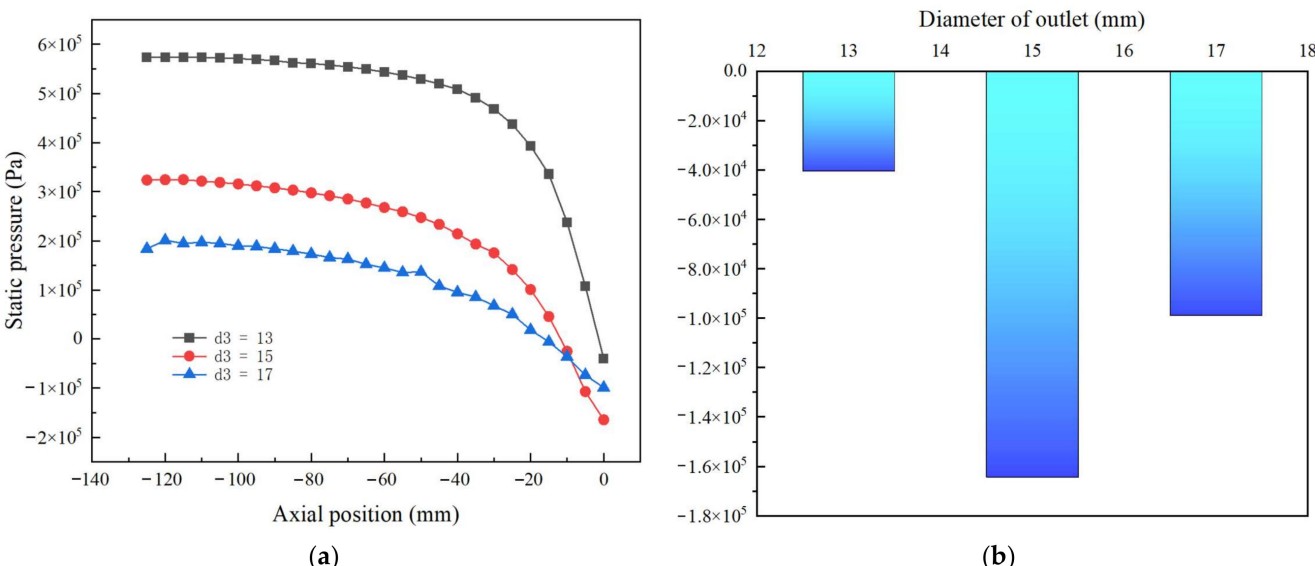

**Figure 10.** Axial static pressure distribution curve (**a**) and minimum static pressure (**b**) of the outlet.

### 3.4. Numerical Simulation Results

The structure of the tangential vortex micro-nano bubble generator was optimized by numerical simulation. The simulation results showed that the optimized central inlet tower was located 10 mm above the inlet of the variable-diameter accelerated vortex cavitation reaction chamber. Due to the longer flow distance between the fluids, the shear force was stronger, the turbulent dissipation rate of the flow field was higher, and the distribution range was wider at this time. The optimized variable diameter contraction angle was 16°. At this moment, the independent fluids in the cavitation reaction chamber collide with each other during the mixing process and produced strong shear force, which was more conducive to the generation of small bubbles. The strong vortex significantly increased the turbulent kinetic energy of the mixed phase, the instability of the gas–liquid interface increased, the surface tension of the bubble decreased, and it was easier to break into the micro-nano level. The optimized generator outlet diameter was 15 mm. At this time, the minimum static pressure at the outlet was the lowest, which was $-1.644 \times 10^5$ Pa. The cavitation formation, development, and collapse of the gas in the liquid were intensified, and the cavitation effect was the most significant.

### 3.5. Bubble Size Measurement Results

The results of this study were based on the experimental method designed in Section 2.5 to test the performance of the optimized micro-nano bubbles. The results of bubble size measurement can be seen in Figure 11a, where the bubbles are spherical and uniform in size, and the bubble size distribution is shown in Figure 11b.

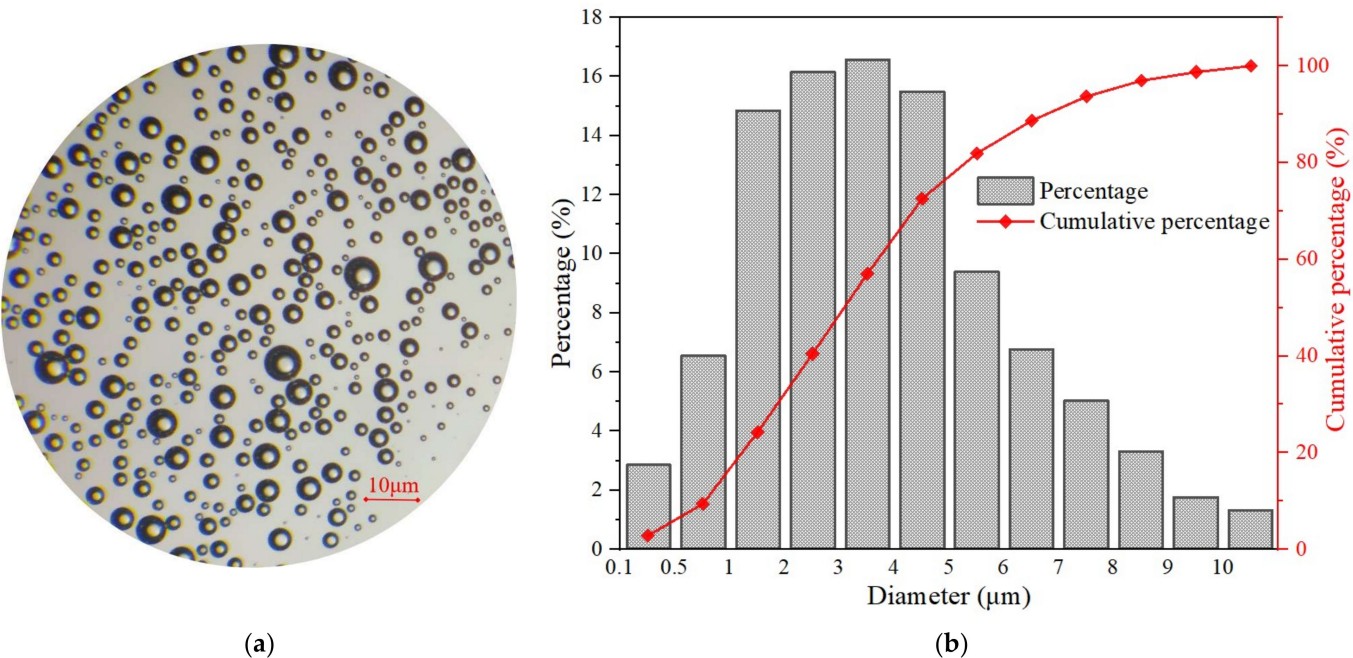

(**a**)                                                                                          (**b**)

**Figure 11.** Micro-nano bubbles under a high-magnification microscope (**a**) and the size distribution of micro-nano bubbles (**b**).

In the range of 1–5 μm, the bubble size accounted for large fractions, accounting for up to 63.1%. Nearly 75% of the bubble diameters were less than 5 μm, and about 25% of the bubble diameters on the right side were in the range of 5–10 μm. The average diameter and minimum diameter of the micro-nano bubbles produced in this work were 3.42 μm and 0.31 μm, respectively. The results of the minimum bubble diameter measured here were basically consistent with the results of the turbulence change rate obtained by the numerical simulation in Section 3.1, which brought in the equation 9 calculation. The bubble test results and the numerical simulation results were mutually verified, which

proved the accuracy of the numerical simulation data and the rationality of the bubble test method. In addition, the bubble size $d_{75}$ of the bubble generator in this study was 5.31 μm, indicating that most of the bubbles were at the micro-nano level. The experimental results showed that the swirl flow bubble generator in this study could produce micro-nano bubbles with smaller sizes and a more uniform distribution. The reason for this phenomenon can be explained, as the swirling effect significantly enhances the turbulence intensity of the liquid and produces a stronger shear force on the bubble, resulting in a stronger bubble rupture process and a greatly reduced bubble diameter. Once the bubble enters the variable-diameter accelerated vortex cavitation reaction chamber, it will undergo severe deformation and then rupture [28].

## 4. Conclusions

A superior tangential swirl micro-nano bubble generation strategy was proposed, which combined the vortex supply chamber and the variable-diameter accelerated vortex cavitation reaction chamber to obtain a stable, high-speed tangential liquid flow while improving the cavitation effect inside the generator. The central air intake column was innovatively installed above the cavitation reaction chamber, which prolonged the shear fracture time of bubbles under high shear force and improved the gas–liquid contact and mixing efficiency. The structure of the tangential vortex micro-nano bubble generator was optimized by numerical simulation. The simulation results showed that the optimized central air intake column was located 10 mm above the inlet of the variable-diameter accelerated vortex cavitation reaction chamber, the optimized variable-diameter contraction angle was 16°, and the optimized generator outlet diameter was 15 mm. In addition, the bubbles generated by the swirl flow micro-nano bubble generator reached the micro-nano level, and the minimum size and average size of the micro-nano bubbles were 3.42 μm and 0.31 μm, respectively. In this study, the characteristics of high efficiency and low power consumption were the characteristics of the swirl flow micro-nano bubble generator introduced in this research. The enforcement of the research provided theoretical guidance and data support for the development of efficient micro-nano bubble generators.

**Author Contributions:** Conceptualization, B.Z. and C.W.; methodology, B.Z.; software, M.X.; validation, X.H., X.X. and M.X.; formal analysis, X.H.; investigation, X.Z.; resources, X.Z.; data curation, M.X.; writing—original draft preparation, X.X.; writing—review and editing, X.H.; visualization, X.X.; supervision, C.W.; project administration, B.Z.; funding acquisition, X.Z. All authors have read and agreed to the published version of the manuscript.

**Funding:** This research was funded by the Jiangsu Water Conservancy Science and Technology Project (2021039) and the Changzhou Sci&Tech Program (China, No. CJ20220025).

**Institutional Review Board Statement:** Not applicable.

**Informed Consent Statement:** Not applicable.

**Data Availability Statement:** Not applicable.

**Conflicts of Interest:** The authors declare no conflict of interest.

## Nomenclature

| | | | |
|---|---|---|---|
| $p$ | Gas phase | $\dot{m}_{pq}$ | Mass transfer from gas phase to liquid phase |
| $q$ | Liquid phase | $R_{pq}$ | The interaction between two phases |
| $\rho$ | Density | $F_q$ | The external volumetric force |
| $\alpha$ | Volume fraction | $F_{lift,q}$ | The lift force |
| $v$ | Fluid velocity | $F_{vm,q}$ | The virtual mass force |
| $g$ | Acceleration of gravity | $v_{qp}$ | The relative velocity between the gas phase and liquid phase |
| $\bar{\bar{\tau}}$ | Stress tensor | | |

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
