# Peer review of "Numerical Simulation and Structural Optimization of Swirl Flow Micro-Nano Bubble Generator"

_coatings, doi:10.3390/coatings13081468_

Round 1

Reviewer 1 Report

This paper deals with numerical simulations of swirl flow in a bubble generator of micrometric size. If I have understood, the idea of the paper is to optimize bubble production by varying some model parameters. Experimental results are also presented. In Section 2,2 the authors write down the basic RANS equations for this two-phase flow coupled to the realizable k-epsilon model for turbulence. I think the paper could be considered for publication in Coatings once the authors respond to the following comments and suggestions:

1) Evidently there is no a mesh-independence test for this model, and I wonder how the authors came to choose a resolution consisting of about 3 million tetrahedral elements. Is this resolution enough to reproduce the experimental data? Please explain.

2) Has Figure 5 an experimental counterpart? Same question for Figure 6(b).

3) Perhaps Figure 7 could some way compared with some experimental data. I think that any numerical result must be validated and validation must be trusted once it reproduces the experiment.

4) In Section 3.4 the authors discuss the numerical results. I am a bit confused. Do the authors mean that optimization in their numerical experiments is achieved for the cases where cavitation is enhanced? Is that the case? I imagine that this finding has some experimental support.

5) Figure 11 is discussed in Section 3.5. I guess that this figure corresponds to experimental results. Please confirm.

I think that the quality of English is acceptable.

Reviewer 2 Report

This paper focuses on the development of a specific type of micro-nano bubble generator using swirling flow and numerical simulation. It does not cover other methods or technologies for generating micro-nano bubbles, nor does it compare the performance of this generator to other existing generators. Therefore, the limitations of this manuscript may include its narrow focus on a specific type of generator and its lack of comparison to other methods or technologies.

The paper does not explicitly compare its research with contemporary research in the same domain. However, it does mention that several typical micro-nano bubble generators based on hydrodynamic cavitation have been developed in the past few decades, such as pressurized dissolution-type, Venturi type, and ejector type. The paper proposes a new type of swirling micro-nano bubble generation strategy that combines the vortex supply chamber and the variable diameter accelerated vortex cavitation reaction chamber to obtain a stable high-speed tangential liquid flow while improving the cavitation effect inside the generator. The paper also provides numerical simulation results and performance test results of the manufactured generator. Therefore, while the paper does not directly compare its research with contemporary research, it does contribute to the development of a new type of micro-nano bubble generator and provides data support for its performance.

Minor English Correction required

Reviewer 3 Report

The article investigates the superior tangential swirl micro-nano bubble generation strategy combining the vortex supply chamber and the variable-diameter accelerated vortex cavitation reaction chamber to obtain a stable high-speed tangential liquid flow while improving the cavitation effect inside a generator. My comments are as follows:

- A comparison of the introduced study with existing experimental data should be represented.

- Employed software for numerical solution of model should be mentioned.

- The accuracy of the obtained solutions for the model should be discussed.

- Some abbreviations are misplaced in text. Please mention the word/definition in full for the first time followed by the abbreviation. Then continue using the abbreviation along the manuscript.

- Minor proofreading is to be considered or run a grammar check on the whole document.

- For enhancing the introduction section with the new publications, old references may be replaced with new ones such as:

Versatile response of a Sutterby nanofluid under activation energy: hyperthermia therapy

Assorted kerosene-based nanofluid across a dual-zone vertical annulus with electroosmosis

Unraveling the nature of nano-diamonds and silica in a catheterized tapered artery: highlights into hydrophilic traits

Scientific Breakdown of a Ferromagnetic Nanofluid in Hemodynamics: Enhanced Therapeutic Approach

good

Round 2

Reviewer 1 Report

I noticed that the authors have responded positively to most of my suggestions and comments. I think the paper looks much better than in its original form. Therefore, I recommend publication of the paper in Coatings.